# Non-Classical Effects of FGF23: Molecular and Clinical Features

**DOI:** 10.3390/ijms25094875

**Published:** 2024-04-30

**Authors:** Luis Martínez-Heredia, Juan Manuel Canelo-Moreno, Beatriz García-Fontana, Manuel Muñoz-Torres

**Affiliations:** 1Instituto de Investigación Biosanitaria de Granada, 18014 Granada, Spain; luismh95@gmail.com; 2Biomedical Research Network in Fragility and Healthy Aging (CIBERFES), Instituto de Salud Carlos III, 28029 Madrid, Spain; 3Endocrinology and Nutrition Unit, University Hospital Juan Ramón Jiménez, 21005 Huelva, Spain; jmccanelo@gmail.com; 4Endocrinology and Nutrition Unit, University Hospital Clínico San Cecilio, 18016 Granada, Spain; 5Department of Cell Biology, University of Granada, 18016 Granada, Spain; 6Department of Medicine, University of Granada, 18016 Granada, Spain

**Keywords:** FGF23, Klotho, calcineurin pathway, chronic kidney disease, XLH, burosumab

## Abstract

This article reviews the role of fibroblast growth factor 23 (FGF23) protein in phosphate metabolism, highlighting its regulation of vitamin D, parathyroid hormone, and bone metabolism. Although it was traditionally thought that phosphate–calcium homeostasis was controlled exclusively by parathyroid hormone (PTH) and calcitriol, pathophysiological studies revealed the influence of FGF23. This protein, expressed mainly in bone, inhibits the renal reabsorption of phosphate and calcitriol formation, mediated by the α-klotho co-receptor. In addition to its role in phosphate metabolism, FGF23 exhibits pleiotropic effects in non-renal systems such as the cardiovascular, immune, and metabolic systems, including the regulation of gene expression and cardiac fibrosis. Although it has been proposed as a biomarker and therapeutic target, the inhibition of FGF23 poses challenges due to its potential side effects. However, the approval of drugs such as burosumab represents a milestone in the treatment of FGF23-related diseases.

## 1. Introduction

Fibroblast growth factor 23 (FGF23) is a protein belonging to the fibroblast growth factor family, and is responsible for regulating phosphorus metabolism [1]. For a long time, it was believed that phosphate–calcium homeostasis was exclusively under the influence of parathyroid hormone (PTH) and 1.25-dihydroxyvitamin D3 (calcitriol) as shown in Figure 1. However, pathophysiological studies with phosphate restriction revealed the existence of other factors involved in the regulation of calcitriol and phosphate levels [2].

FGF23 consists of 251 amino acids and is mainly expressed in bone, specifically in osteoblasts and osteocytes [3], although it can also be detected in various organs such as the liver [4], brain [5], heart [6], thyroid [7], intestine [8], and skeletal muscle [9].

FGF23 acts as a potent inhibitor of renal phosphate reabsorption, inducing renal phosphate excretion and decreasing the surface expression of the sodium-associated phosphate transporters NaPi-IIa and NaPi-IIc, which are located in the proximal tubule [10]. In addition, at the renal level, FGF23 decreases the expression of cytochrome P450 family 27 subfamily B member 1 (CYP27B1), the key enzyme for the formation of calcitriol [11]. These effects are mediated by the α-klotho co-receptor.

Although the effects on phosphate metabolism are well established, recent research has revealed the pleiotropic effects of the hormone on other physiological systems beyond the renal system [12]. At the bone level, an increasing number of preclinical studies infer local actions of FGF23 on bone remodeling, as well as its relationship in a clinical context with bone fragility and fracture risk in patients with or without kidney disease [13,14,15]. On the other hand, FGF23 has been recognized as an important biomarker at the cardiovascular level [16,17]. In addition, it appears to be involved in iron metabolism [18], immune response [19], or glucose metabolism [20]. In this article, we will explore some of the recently discovered non-classical effects of FGF23.

## 2. FGF23 Effects over Vitamin D, PTH, and Bone Metabolism

### 2.1. Vitamin D

The best-known effect of FGF23 on vitamin D metabolism is the decrease in calcitriol on its passage at the renal level by α1-hydroxylase in an α-klotho-dependent, MAP kinase-mediated mechanism [21]. Knockout mice for FGF23, α-klotho, or both exhibit higher levels of calcitriol and α1-hydroxylase [21]. Furthermore, the infusion of FGF23 allowed a reversal of hypervitaminosis in FGF23 knockout mice [22]. This phenotype is clinically similar in patients with loss-of-function mutations in FGF23, resulting in hyperphosphoremia, hypervitaminosis D, and calcinosis [23,24]. On the other hand, FGF23 induces the inactivation of calcitriol through an increased expression of 25-Hydroxyvitamin D-24-hydroxylase (CYP24A1) [25].

In turn, calcitriol could regulate the expression of FGF23. It has been observed in patients that paricalcitol supplementation potentiated the production of FGF23 in a mechanism mediated by the vitamin D receptor (VDR) [26]. In mice, the genetic inactivation of the VDR reduced the levels of active FGF23 compared to wild-type mice, which implies the need for this receptor for vitamin D to mediate its effects on FGF23 [27]. Supporting this concept, α1-hydroxylase knockout mice with functioning VDR resulted in a phenotype with decreased FGF23, which increased with calcitriol administration [28]. In contrast, the intraperitoneal injection of paricalcitol in vitamin D-deficient rats (via standard vitamin D-deficient diet) caused FGF23 reduction in a vitamin D-mediated effect and PTH reduction [29]. In humans, there is a discrepancy in these facts. A meta-analysis of randomized placebo-controlled clinical trials found that vitamin D administration was associated with an increase in circulating FGF23 levels in a dose-dependent manner [30]. However, another meta-analysis of randomized clinical trials did not confirm these findings [31]. The discrepancy in the two studies could be mediated by the different forms of vitamin D administration in the form of calcitriol or 25-hydroxyvitamin D3 (calcifediol) [32].

### 2.2. PTH

FGF23 is a negative regulator of PTH mRNA expression and secretion and can be stimulated by non-oxidized PTH, particularly in the context of chronic kidney disease (CKD) [33]. The relationship between FGF23 and PTH is further influenced by dietary phosphate intake, with FGF23 levels decreasing on a low-phosphate diet and increasing with a high-phosphate diet [34]. These findings suggest a complex interplay between FGF23 and PTH in the context of CKD and dietary phosphate intake.

The relationship between PTH and FGF23 is fundamental to the regulation of phosphate–calcium metabolism in the body. Both hormones play key roles in maintaining adequate levels of phosphate and calcium in the body. Under normal physiological conditions, there is an inverse relationship between PTH and FGF23 levels [35]. When FGF23 levels increase, they tend to inhibit PTH release. This effect of FGF23 would be carried out through two pathways: one mediated by the activation of the ERK1/2 pathway in an α-klotho-dependent manner and another in a klotho-independent manner through NAFT/calcineurin [36,37]. In addition, it has been observed that FGF23 can increase the presence of vitamin D receptors (VDR) and calcium-sensitive receptors in parathyroid cells, which could pathophysiologically decrease PTH expression [13].

In advanced kidney disease (CKD), FGF23 loses its ability to suppress PTH [13]. Prolonged exposure to FGF23 may even increase PTH secretion. In CKD, a state of hyperparathyroidism develops despite the increased and possibly inhibitory effect of FGF23. In this uremic condition, chronic exposure to FGF23 seems to lead to the down-regulation of Klotho and FGFR1c receptors on parathyroid cells, generating resistance to FGF23 [38]. Furthermore, in the clinical situation of hyperparathyroidism in CKD, extremely high levels of FGF23 decrease calcitriol, which in turn increases PTH levels. Resistance to FGF23 would result in increased PTH, increased phosphate, and decreased VDR.

At the bone level, there is conflicting evidence on the effect of PTH on FGF23 release. In murine models, PTH administration after parathyroidectomy resulted in increased FGF23 expression [39,40], whereas other studies showed the opposite effect, with reduced FGF23 after PTH administration [41,42]. In human models, PTH infusion increased FGF23, whereas supraphysiological PTH infusion reduced levels. These contradictory results could be due to other factors, such as the hypophosphatemic effect of PTH, relative elevation of calcitriol, and variations in calcium levels [43]. In another study, in patients with hypoparathyroidism, an increase in FGF23 was observed without changes in calcitriol concentration [44].

### 2.3. Bone Metabolism

In vitro studies with murine osteoblasts have shown that an increase in FGF23 leads to an increase in bone mineralization, with a dose-dependent increase in osteocalcin, osteopontin, and alkaline phosphatase, and this effect is independent of calciferol and Klotho [45]. However, the specific effects of FGF23 on osteoblast differentiation and mineralization are still debated since others have suggested that FGF23 and soluble Klotho can inhibit mineralization and osteoblast activity [46,47]. In CKD rats there appears to be an inhibition of the Wnt/βcatenin pathway in osteoblasts [48]. These indications support an auto/paracrine effect of FGF23 on the bone matrix [49]. However, a supraphysiological concentration of FGF23 inhibited the effect of alkaline phosphatase by increasing extracellular pyrophosphate, decreasing inorganic phosphate with a consequent decrease in bone mineralization [50]. Therefore, the effect of FGF23 could be different depending on FGF23 concentrations [49].

The absence of VDR decreases FGF23 levels, causing a decrease in osteoclastogenesis. Thus, FGF23 may be involved in the RANKL signaling pathway [51]. Another study with human monocyte-derived osteoclasts showed a biphasic response to FGF23 with an inhibition of the early stages of osteoclastogenesis followed by an increase in osteoclastic activity [52].

Therefore, FGF23 could affect bone remodeling through its effects on bone mineralization and osteoclastogenesis. According to some of the effects described, FGF23 could increase bone resorption and therefore could lead to deleterious effects on bone. The association between FGF23 and bone mass is contradictory. In a study of 3014 Swedish men with osteoporosis, FGF23 levels showed a positive association with bone mass density (BMD), but no correlation was found after adjustment for confounding factors [53]. In another study of 5994 men with osteoporosis there was a similar neutral effect on femoral neck BMD [54]. In contrast, in studies in postmenopausal women involving cohorts of 60 and 123 patients, there was a negative relationship between FGF23 and BMD [55,56]. This detrimental effect in postmenopausal women of FGF23 on bone was reinforced in another study where it was associated with a deterioration of bone trabecular microarchitecture [57]. Therefore, there is insufficient evidence to determine the role of FGF23 in the pathogenesis of osteoporosis to date.

Despite contradictory evidence on BMD, increased levels of FGF23 could be a predictor of fracture risk both in the population without CKD [58,59] and with or without end-stage CKD [54,59,60]. FGF23 could play a role in the pathogenesis of bone disease in CKD. In CKD, FGF23 does not inhibit PTH production and decreases calcitriol levels, leading to further hypocalcemia and the generation of secondary hyperparathyroidism. In addition, excess FGF23 could inhibit bone mineralization by reducing alkaline phosphatase and pyrophosphate accumulation [61]. In animal models of CKD, the administration of antibodies against FGF23 resulted in improved bone quality [62,63]. However, in other studies with animal models this neutralization of FGF23 resulted in an exacerbation of hyperphosphatemia, increased calcitriol, and increased arterial calcification [49].

## 3. Cardiovascular Effects

A number of studies have demonstrated the association between increased mortality and increased levels of FGF23 independently of the presence of CKD, so the increased mortality could be due to a greater increase in cardiovascular risk [64,65]. In a recent meta-analysis, Menglu Liu et al. show the relationship between FGF23 levels and myocardial infarction, stroke, heart failure, and cardiovascular mortality [66].

### 3.1. Endothelial Dysfunction

The results in different in vivo and ex vivo studies seem contradictory. Lindberg et al. found no effect of FGF23 on endothelial function [67]; however, in other studies, they state that FGF23 results in aortic endothelial dysfunction.

From a clinical perspective, elevated serum levels of FGF23 are associated with endothelial dysfunction both in patients with stage 3–4 CKD [68] and in the general population [69]. Similarly, after renal transplantation and the normalization of FGF23 levels, there is an improvement in endothelial function [70].

### 3.2. Left Ventricular Hypertrophy

In clinical practice, an association between increased FGF23 levels and left ventricular hypertrophy (LVH) has been observed independently of renal involvement [71], both in patients with normal FGF23 levels and no CKD [72], and in those with increased FGF23 levels in the context of CKD [73]. This association has been confirmed experimentally, where increased FGF23 on cardiomyocytes results in a change in gene expression leading to myocardial hypertrophy. This effect has been replicated in mice in the absence of klotho, confirming that the mechanism operates in a klotho-independent manner [73]. This process appears to be mediated by the activation of the fibroblast growth factor receptor 4 (FGFR4) and the PLCgamma/calcineurin/NFAT pathway in cardiomyocytes [74].

Conversely, other authors have not found these effects at the experimental level and argue that myocardial damage is not due to the direct effect of FGF23 [30,75,76]. In addition, it has been observed that mice and children with X-linked hypophosphatemia (XLH) do not present with LVH [77,78,79]. These studies suggest that phosphate may play an important role in the cardiotoxic function of FGF23.

There is evidence of a paracrine effect of FGF23 in cardiomyocytes themselves. It has been demonstrated that cardiomyocytes are capable of expressing FGF23 [80]. In clinical studies, the presence of FGF23 has been observed in heart explants from patients with ischemic or dilated heart disease in whom cardiac transplantation has taken place [81]. In parallel, an increase in FGF23 has been observed at both bone and cardiac levels in mice with myocardial infarction [82] and this increase in FGF23 in cardiac pathological situations could also be observed in a model of LVH with aortic constriction in which an increase in FGF23 expression at both cardiac and bone levels was determined [83]. These results are consistent with those obtained at the clinical level where patients who required intra-aortic balloon pumps exhibited higher levels of FGF23, which were associated with increased mortality [84].

### 3.3. Cardiac Fibrosis

Fibrosis is an adaptive response to tissue damage, although excessive deposition in the extracellular matrix leads to architectural disruption and damage to healthy tissue [85]. Although FGF23 has a clear effect on the myocardium, its effect on cardiac fibrosis is not clear. FGF23 could induce myocardial fibrosis through the activation of β-catenin and TGF-β1. In adult cardiac fibroblasts, FGF23 induces, in a dose-dependent manner, the overexpression of profibrotic genes such as β-catenin and procollagen I and II [86]. In addition, FGF23 increases the expression of TGF-β1 and procollagen I, along with the proliferation of neonatal rat cardiofibroblasts. This effect on TGF-β1 could be mediated by fibroblast growth factor 1 (FGFR1) and the paracrine secretion of FGF23 since the blockade of FGFR1 prevents fibrosis [87]. These effects have also been observed in vivo in klotho knockout mouse models with FGF23 elevation, where fibrosis developed in a mechanism dependent on increased procollagen I and TGF-β1 [88]. The injection of FGF23 by adenovirus in mice demonstrated a significant increase in cardiac fibrosis, the detrimental effect of which was prevented by an inhibition of β-catenin. Thus, in the first instance, FGF23 could have a benefit in healing after a cardiac ischemic event as it stimulates fibroblasts for the initial development of fibrosis [89], but prolonged exposure to high levels of FGF23 could result in pathological fibrosis [90].

### 3.4. FGF23 and the Renin–Angiotensin–Aldosterone System

The activation of the renin–angiotensin–aldosterone system (RAAS) is another recognized factor in the fibrosis, hypertrophy, and inflammation of the myocardium [91]. In this system, renin is converted into angiotensin I and into angiotensin II by the angiotensin-converting enzyme (ACE), which is the factor that conditions fibrosis and hypertrophy. Angiotensin II is metabolized by angiotensin-converting enzyme II (ACE2), whose products have vasodilatory and hypotensive effects. FGF23 has been shown to activate the RAAS by inhibiting ACE2 [92]. On the other hand, the RAAS can induce FGF23 expression by both angiotensin II and aldosterone [93], whereas calcitriol suppresses renin and thus the RAAS [94]. It could be hypothesized that FGF23 inhibits the formation of calcitriol, indirectly mediating in the RAAS. In conclusion, although the relationship between FGF23 and the RAAS is not clearly described, FGF23 could affect fibroblasts and cardiomyocytes through a non-classical activation of the RAAS [95].

### 3.5. Atherosclerosis

Observational studies have noted an association between FGF23 levels and the presence of vascular calcifications [96,97]. FGF23 is expressed in carotid arteries with atheromatous plaques and in the coronary arteries of transplant patients [98,99]. However, it has not been determined whether FGF23 is a cause or a consequence of endothelial injury [98]. In vitro results present contradictory positive and negative results regarding the formation of intravascular calcification [100,101]. While the overexpression of FGF23 in rodents does not generate vascular calcification [102], FGF23 deficiency results in severe calcifications [103]. Therefore, the role of FGF23 on atherosclerosis has not been reliably determined.

## 4. FGF23 and Iron Metabolism

### 4.1. Iron Deficiency

Research on the interaction between FGF23 and iron metabolism originated from clinical observations in patients with autosomal dominant hypophosphatemic rickets (ADHR) and iron deficiency. It was noted that many female carriers of the disease had no associated clinical symptoms, until menarche, related to iron deficiency [104]. To explore this relationship between FGF23 and iron, it was observed that low iron levels correlated with elevated levels of the inactive form of FGF23 (cFGF23), after the cleavage of the active form (iFGF23), in both the ADHR population and healthy controls. This negative correlation was maintained in patients with ADHR and iFGF23, but not in healthy controls who maintained normal iFGF23 values [105].

These findings were confirmed using ADHR mice, where increased bone FGF23 mRNA and increased serum cFGF23 levels were observed in both affected mice and controls. In contrast, ADHR mice showed higher iFGF23 levels and hypophosphatemia during iron deficiency, whereas wild-type controls maintained normal iFGF23 and phosphorus levels [106]. Therefore, these data support the idea that iron deficiency can stimulate both the transcriptional and post-transcriptional activity of FGF23, maintaining normal iFGF23 levels in healthy patients [107].

In chronic kidney disease (CKD), alterations in FGF23 cleavage mechanisms are also observed, although the underlying mechanisms are unknown. In CKD, both absolute iron deficiency, characterized by a decrease in total iron, and relative iron deficiency, which is due to a reduced ability to utilize circulating iron from its stores, are present [108]. This iron deficiency, either absolute or relative, has been associated with increased levels of the active form of FGF23 (iFGF23) in animal models with CKD [109]. In addition, elevated levels of FGF23 have been observed in patients with CKD and iron deficiency [110]. Therefore, iron deficiency in CKD patients could contribute to the increase in FGF23 [111].

In situations of acute anemia, an increase in cFGF23 levels without significant changes in iFGF23 has also been observed, both in animal models and in patients requiring red blood cell transfusions in intensive care. In the animal model, these effects appear to be mediated by the reduction in the enzyme polypeptide N-acetylgalactosaminyltransferase 3, which prevents FGF23 cleavage by keeping cFGF23 and iFGF23 levels balanced [112].

### 4.2. Hypoxia

There are no clear mechanisms by which iron deficiency appears to increase FGF transcription [30]. One of the proposed mechanisms is the activation of hypoxia-induced factors (HIF). It has been observed that increased HIF1α resulted in increased FGF23 transcription by binding to its promoter in osteoblast cultures [113]. In this regard, iron-deficient mice with HIF1α blockade modified FGF23 levels upward. Therefore, it is suggested that HIF1α may be one of the mediators of increased FGF23 in iron deficiency, anemia, or inflammation even though HIF1α does not appear to be a necessary factor for FGF23 expression since the knockout of HIF1α in mature osteoblasts does not affect FGF23 levels in wild-type mice or those with XLH [114].

### 4.3. Erythropoietin (EPO)

EPO stimulates erythropoiesis by stimulating erythroblast differentiation, survival, and proliferation [115]. Additionally, it ameliorates iron deficiency by reducing hepcidin through the release of iron from intracellular compartments [116]. Similar to iron deficiency, both endogenous and exogenous EPO increases transcription through the cleavage of FGF23 [117,118]. Furthermore, patients with mutations that increase EPO production have increased levels of cFGF23 without changes in iFGF23 [119]. In patients with CKD, the acute administration of EPO also results in modest increases in iFGF23 relative to cFGF23 [117,118]. Furthermore, this EPO-mediated increase in FGF23 is due to increased mRNA expression in bone marrow hematopoietic precursors [117,120]. However, up to 40% of the increase in cFGF23 is maintained after bone marrow ablation. This could be due to a possible secretion of cFGF23 by osteoblasts [121].

### 4.4. Iron Repletion

By correcting iron deficiency, cFGF23 levels are reduced, due to a decrease in its transcription, while in healthy patients without CKD iFGF23 levels do not change with iron repletion; in CKD patients, one would expect that restoring iron levels would result in a reduction in iFGF23, although this effect is not fully confirmed [122,123]. A clinical trial in hemodialysis patients showed that the restoration of iron levels by oral iron intake decreased iFGF23 levels from baseline [110]. In parallel, the same effect was observed in CKD mice given iron [124]. An analogous effect was achieved in the ADHR model, where there was also an increase in iFGF23 due to transcriptional upregulation and the impossibility of post-transcriptional cleavage. ADHR mice with iron deficiency exhibited normal levels of iFGF23 and phosphorus after the restoration of iron levels [125]. This effect also appeared to translate to ADHR patients, where a case is reported with a cessation of symptomatology after IV iron administration, and in a clinical trial where iron supplementation reduced iFGF23 levels and hypophosphatemia [107].

Paradoxically, cases of hypophosphatemia have been reported after intravenous IV iron administration [18]. In a clinical trial with 55 women using dextran iron versus carboxymaltose iron, cFGF23 levels were shown to be reduced in both groups, while the carboxymaltose group increased iFGF23 and half of the patients developed hypophosphatemia [122]. This effect was further corroborated in a clinical trial of 2000 patients with anemias of different origins [123]. The mechanism by which iron carboxymaltose produces an increase in iFGF23 may be similar to the ADHR model. When these phosphaturic preparations are administered in the situation of iron deficiency with increased FGF23 transcription, they would inhibit the FGF23 cleavage process leading to an increase in iFGF23. It is proposed that the sugar compounds in the preparation could increase the glycation processes that prevent the cleavage of the molecule [122].

## 5. FGF23 and Renal Failure

As previously described, there is a close association between the levels of FGF23 and CKD. Although its value as a clinical biomarker remains under discussion, some studies showed that FGF23 is an independent predictor of adverse outcomes in CKD patients, observing an association between high FGF23 levels with increased cardiovascular risk and progression to end-stage renal disease requiring renal replacement therapy (RRT) [126,127,128,129]. In an observational study of 180 healthy adults and 18 adults with stage 3–5 CKD, Smith et al. determined that there was a high variability in FGF23 measures, specifically iFGF23, as a diagnostic tool, suggesting that risk-related thresholds of cFGF23 measures may be more appropriate for clinical decision-making [130]. In contrast, another study conducted in 2544 Canadian patients with CKD estimated that the measurement of new biomarkers such as FGF23 does not imply an improvement in the prediction of RTT when added to conventional risk factors; however, it improved the prediction of death at one year [131].

Regarding acute kidney injury (AKI), several studies indicate that there is a significant increase in FGF23, postulating it as a promising early biomarker of AKI [132,133,134]. Comparative studies between mice and humans show consistent results in the increase in FGF23 related to AKI after cardiac surgery [132]. Compared with other cardiac surgeries, plasma FGF23 levels were consistently higher in those who developed AKI than those who did not; specifically, cFGF23 levels were more robust and significantly associated with an increased risk of severe AKI and the need for RRT or death [133]. To date, the mechanisms by which FGF23 increases after the onset of AKI remain unknown. However, a study in murine models observed a decrease in the severity of AKI induced by ischemia–reperfusion in mice pretreated with FGF23, suggesting a protective role of FGF23 in AKI, and the promotion of renal tubular regeneration and vascular repair [135]. Thus, FGF23 alterations in AKI highlight the importance of the bone–kidney–heart axis

## 6. Inflammation and Immune System

FGF23 stimulates the secretion of proinflammatory cytokines. In an animal model, elevated levels of FGF23 in mice increased serum and hepatic levels of C-reactive protein (CRP) and interleukin-6 (IL-6) [136]. This increase in inflammatory markers can also be observed in patients with moderate CKD the higher the level of FGF23 [137]. The effect seems to be mediated by fibroblast growth factor receptor 4 (FGFR4), since its blockade in nephrectomized mice with elevated FGF23 resulted in a reduction in inflammatory markers. Similarly, inhibition with cFGF23 reduces inflammation and improves renal function in a mouse model of diabetic nephropathy [138]. In a feedback loop, different proinflammatory cytokines such as TNFα, IL-β1, and IL-6 have been shown to increase the expression of FGF23 [109,139,140,141]. This mechanism would be similar to that previously presented for iron deficiency and EPO, with increased transcription and, at the same time, post-transcriptional excision, but which in situations with an alteration of this control, such as CKD, would give rise to increased iFGF23 [109]. In patients with CKD, higher levels of FGF23 are associated with increased infection-related mortality [142] and hospitalizations for infection [143]. In addition, FGF23 has been shown to dysregulate the immune response in mouse models, altering the leukocyte response and bacterial clearance. This effect could be reversed by antibodies against FGF23 [144].

## 7. FGF23 and Obesity

The relationship between obesity and FGF23 levels is not completely clarified. In the last decade, the possibility has arisen that the fibroblast growth factor FGF23 may be linked to fat content and alterations in lipid and glucose metabolism. One study has observed elevated serum levels of FGF23 in individuals with obesity, especially those with abdominal obesity. Another study conducted in 1599 normoglycemic individuals, including men and premenopausal and postmenopausal women, found independent associations between the presence of abdominal obesity and increased serum levels of FGF23 in specific groups. This suggests that serum FGF23 levels could indicate the risk of metabolic and cardiovascular disease in men and postmenopausal women [145]. In contrast, another study in obese young people with or without insulin resistance observed that the group of patients with insulin resistance had lower levels of FGF23 [146]. However, a study by Hanks and colleagues with 1040 participants reported a positive association between insulin resistance and FGF23 levels in the absence of chronic kidney disease. Furthermore, they found a positive relationship between resistin levels and FGF23, without observing an association with adiponectin [147]. Figure 2 schematizes the endocrine effects of FGF23 described above.

The mechanism by which adipokines regulate FGF23 secretion is not completely understood. However, some studies have observed that adiponectin and leptin impact FGF23 secretion in different ways. High levels of adiponectin reduce renal Klotho secretion and FGF23 production by osteocytes. This mechanism is mediated by the renal adipokine receptors ADIPOR1 and ADPR2 [148].

Leptin increases the expression of bone FGF23 but not renal Klotho, which reduces the production of calcitriol and consequently enhances PTH secretion. Additionally, leptin directly stimulates PTH secretion. The combined elevation of FGF23 and PTH leads to increased phosphaturia [149].

## 8. FGF 23 as a Biomarker

There are several hypophosphatemic diseases associated with increased FGF23, such as autosomal dominant and recessive hypophosphatemic rickets/osteomalacia (ADHR, ARHR) [150,151], X-linked hypophosphatemic rickets/osteomalacia (XLH) [152], hypophosphatemic rickets/osteomalacia associated with McCune–Albright syndrome (MAS)/fibrous dysplasia (FD) [153], or epidermal nervous syndrome [154]. Other diseases that cause acquired hypophosphatasemia due to increased levels of FGF23 are tumor-induced rickets/osteomalacia (TIO) [155], hypophosphatemia following intravenous iron infusion [156], and biliary atresia [157].

On the contrary, deficiencies in FGF23 levels, either due to genetic mutations or due to treatments, lead to hyperphosphatemic diseases such as Hyperphosphatemic Tumor Calcinosis [158]. Finally, in CKD, a hyperphosphatemic state occurs despite the high production of FGF23. This particular case could be due to an attempt to compensate for phosphate levels by FGF23 [159].

Due to the wide range of systems in which FGF23 participates, this protein has emerged as a potential biomarker for cardiovascular risk, with high levels associated with adverse outcomes such as heart failure and arrhythmias [90]. It is also a promising marker for identifying high-risk patients in chronic and acute diseases, particularly chronic kidney disease, where it is strongly linked to excess morbidity and mortality [160]. In patients with type 2 diabetes, serum levels are elevated in those with diabetic nephropathy [161]; however, the exact role of FGF23 in these conditions, and whether its association with clinical events is causal, remains a topic of debate [162]. FGF23’s involvement in phosphate–calcium metabolism further underscores its potential as a biomarker for various pathologies such as the risk of bone fractures [163].

## 9. FGF23 as a Therapeutic Target

Conventional therapy for hypophosphatemic rickets adult patients typically involves oral phosphate salts administered twice daily, along with active vitamin D metabolites. The treatment goal is symptom improvement rather than serum phosphate level normalization. A close monitoring of plasma calcium, PTH, creatinine, and 24-h urinary calcium excretion is essential to prevent tertiary hyperparathyroidism due to phosphate overdose and hypercalciuria leading to nephrocalcinosis and renal insufficiency from excessive calcitriol treatment [164,165].

FGF23 has been identified as a potential therapeutic target in several diseases [166]. Excessive levels of FGF23 have been associated with adverse effects, particularly in patients with chronic kidney disease [162]. Targeting FGF23 signaling with antibodies and inhibitors has shown promise in improving disease phenotypes in model mice and clinical trials. However, the specific mechanisms and potential benefits of FGF23 inhibition in CKD patients require further investigation [167]. The loss of FGF23 can generate serious drawbacks such as hypophosphatemia and soft tissue calcifications. Regarding CKD models, an increase in mortality has been observed in preclinical trials using a high-affinity blockade of FGF23 [62]; however, an increase in survival has been observed in end-stage CKD patients using calcimimetics as the inhibition of PTH production results in a modest reduction in FGF23 levels [168].

Burosumab, a low-affinity monoclonal antibody targeting fibroblast growth factor 23, has been approved for the treatment of XLH. Clinical trial results show a normalization of phosphate homeostasis and healing of rickets with a favorable safety profile [169,170].

More recently, studies in adults with XLH and subjects with tumor-induced osteomalacia have also been published with successful results. In both cases, the regulatory agencies have approved this new indication [171,172].

The FGF23-FGF/Klotho receptor pathway has been postulated as a promising pharmacological target for the treatment of phosphate disorders. One of the most successful treatments has been the direct inhibition of FGF23 using monoclonal antibodies (burosumab) [169,170]. Another approach would be the inhibition of FGF23 binding to the Klotho coreceptor by binding small molecules to the glycosyl hydrolase (GS1 and GS2) domains of Klotho [173]. Additionally, there is another strategy that proposes the inhibition of the FGF receptor to increase blood phosphate and calcitriol levels. This strategy is based on the inhibition of downstream signals of the FGF receptor such as the MAPK pathway and it has been tested in solid tumors [174]. While the direct inhibition of FGF23 is used as a treatment in patients, the inhibition of Klotho binding and inhibition of the FGF23 receptor signal are still in the development phase. However, the preclinical studies suggest that there are multiple approaches to inhibit excessive FGF23 activity, thereby addressing FGF23-related hypophosphatemic disorders.

Regarding future therapeutic perspectives, different FGF23 inhibitor compounds are being studied. The C-terminal fragment of FGF23 is generated by proteolytic cleavage at the RXXR motif that acts in the Klotho interaction domain. This peptide has been tested in mice, showing improvements in hematopoietic as well as iron deficiency levels [175]. Furthermore, small molecules based on vHTS affording 1 are being developed with promising results in cell cultures [176]. Finally, in murine models, analogs of ZINC13407541 were discovered to pharmacologically inhibit FGF23. The administration of either of these compounds resulted in the blockade of FGF23 signaling and led to elevated serum phosphate and calcitriol levels in Hyp mice. Furthermore, the long-term parenteral administration of these analogs promoted linear bone growth, enhanced bone mineralization, and narrowed the growth plate in Hyp mice [177].

## 10. Conclusions

In summary, FGF23 plays a pivotal role across various physiological systems, notably in the regulation of phosphorus metabolism. Its impact extends beyond phosphate homeostasis, influencing bone metabolism, cardiovascular health, iron homeostasis, immune response, and metabolic disturbances associated with obesity. The intricate interplay between FGF23, vitamin D, PTH, and bone metabolism underscores its multifaceted involvement in maintaining mineral balance and bone integrity. Furthermore, the FGF23 association with cardiovascular pathologies, such as left ventricular hypertrophy and atherosclerosis, underscores its potential as a biomarker for assessing cardiovascular risk. Despite promising therapeutic implications in targeting FGF23 signaling, further research is imperative to elucidate its precise mechanisms and potential therapeutic benefits, particularly in chronic kidney disease where elevated FGF23 levels contribute to adverse outcomes. Nevertheless, the approval of drugs like burosumab for the treatment of hypophosphatemic rickets signifies a significant advancement in leveraging FGF23 modulation for therapeutic purposes, offering hope for patients afflicted with related disorders.

## Figures and Tables

**Figure 1 ijms-25-04875-f001:**
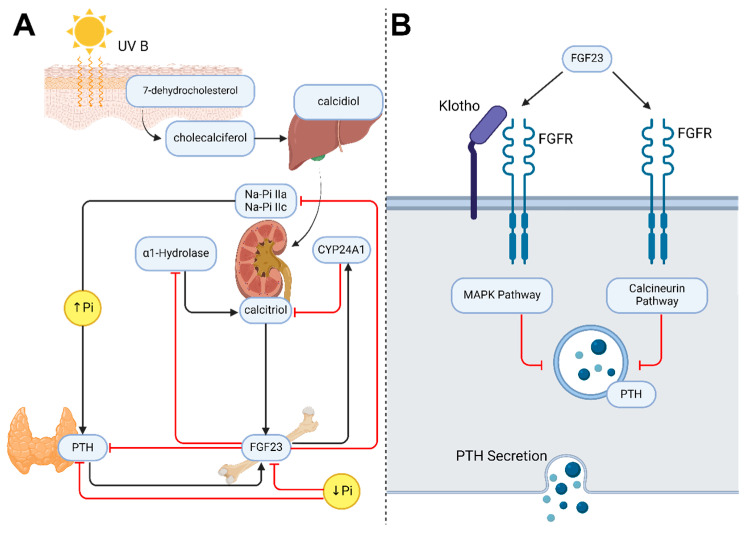
Classical effects of FGF23. (**A**) Mechanism of action of FGF23. (**B**) Signaling pathways of FGF23.

**Figure 2 ijms-25-04875-f002:**
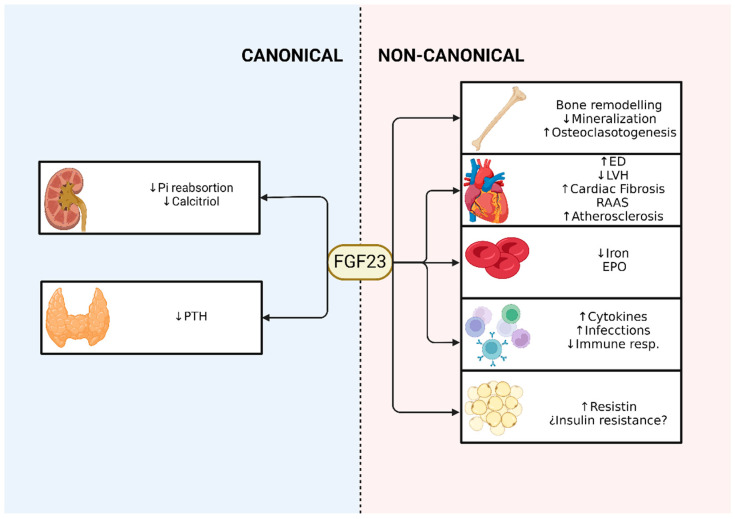
Classical and non-classical effects of FGF23.

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
