# Peer review of "Non-Classical Effects of FGF23: Molecular and Clinical Features"

_ijms, 2024, doi:10.3390/ijms25094875_

Round 1

Reviewer 1 Report

Comments and Suggestions for Authors

This is a very consice review article discussing both the classical (calcium-phosphate homeostasis-related) and non-classical (such as myocardial, renal or iron homeostasis related) effects. Especially in the field of nephrology, increased phosphate levels with elevated FGF23 (and PTH) are clinical issues hard to solve, that have long-term detrimental effects not only on the skeletal system (renal osteodystrophy) but also on the cardiovascular system with increased vascular calficication and "ossification", increasing pulse pressure and myocardial hypertrophy. These important topics are also discussed. 

Comments on the Quality of English Language

Overall the paper is well written, only typos and some sentences need to be corrected for better readability.

For instance:

1) "At the clinical level" is too often used (lines 166 then 171, 193, etc)

2) Sentence in line 182 needs re-phrasing, in present form is hard to understand.

Author Response

Reviewer 1

First, we want to thank you for your effort in reviewing our manuscript and for your constructive comments, which have undoubtedly contributed to improving the quality of our manuscript. Please, find below a report with our point-by-point answers to your comments and attached the manuscript with all the changes highlighted in red in the updated manuscript.

1) "At the clinical level" is too often used (lines 166 then 171, 193, etc)

Thanks for the comment. We have corrected that phrase with synonyms such as in a clinical context with bone fragility and fracture risk…”, “From a clinical perspective, elevated serum levels of FGF23 are associated…”, “In clinical practice, an association between increased FGF23 levels and left ventricular hypertrophy (LVH) has been observed…”

We hope those typos have been eliminated from the article

2) Sentence in line 182 needs re-phrasing, in present form is hard to understand.

Thanks for the suggestion, we have rewritten this sentence as follows ” Besides, it has been observed that mice and children with X-linked hypophosphatemia (XLH) do not present LVH

We hope this makes it clearer.

Reviewer 2 Report

Comments and Suggestions for Authors

Review of the manuscript  ijms-2979773

Non-classical Effects of FGF23: Molecular and Clinical Features

By Luis Martínez-Heredia et al.

The evaluated manuscript is a narrative review discussing selected physiological and pathophysiological roles of fibroblast growth factor 23 (FGF23). In the initial part of the manuscript, the authors describe the "canonical" importance of FGF23 in calcium-phosphate homeostasis, as illustrated in Figure 1. Later in the manuscript, the authors briefly describe the importance of FGF23 in cardiac disorders (such as endothelial dysfunction, left ventricular hypertrophy, and atherosclerosis), iron metabolism, the inflammatory process, and obesity. The final, marginal part of the paper mentions the pharmacological possibility of modulating FGF23 pathways (e.g., burosumab).

The review presents well-thought-out characters and may hold educational value as it summarizes the most important issues related to the role of FGF23 in the body.

However, in my opinion, the manuscript is currently too superficial in several aspects, and the authors should consider making some changes.

1. Figure 1 - The relationships presented are overly simplified. It is worth considering introducing a more detailed vitamin D metabolism pathway into the figure, including biochemical transformations occurring in the skin, hydroxylation in the liver, etc.

2. In the section discussing the characterization of additional, pleiotropic properties of FGF23, it is important to provide a comprehensive description of the role of this protein in the pathogenesis of chronic kidney disease (CKD). This is particularly crucial in the context of the potential diagnostic and prognostic significance of FGF23, as well as in monitoring this protein in CKD patients undergoing renal replacement therapy.

3. The role of FGF23 as a proposed marker of acute kidney injury (AKI) should also be mentioned.

4. The paper would improve in quality by including a section that describes the secretion of FGF23 in hypoxia independently of iron deficiency.

5. Fragment "FGF6 and obesity" – what is the impact of adipokines on FGF23 secretion?

6. It is worth mentioning hypophosphatemic diseases caused by FGF23 overproduction and hyperactivity, as well as hyperphosphatemic disorders.

7. Future therapeutic prospects - Please provide a more detailed description of the potential for modulating the FGF23-FGF receptor/Klotho pathway as a novel drug target for disorders related to phosphate and bone metabolism.

8. Future prospects (continued) – What are the other options for antagonizing FGF23 (beyond the already registered burosumab)?

Author Response

First, we want to thank you for your effort in reviewing our manuscript and for your constructive comments, which have undoubtedly contributed to improving the quality of our manuscript. Please, find below a report with our point-by-point answers to your comments and attached the manuscript with all the changes highlighted in red in the updated manuscript.

Round 2

Reviewer 2 Report

Comments and Suggestions for Authors

Thank you for sending an updated version of the manuscript and for your detailed response to my comments in the previous review.

Authors addressed my suggestions and introduced corrections listed in rebuttall letter .

I believe that the manuscript has gained in quality and is eligible for publication